# Tyrosine Kinase Inhibitors for Glioblastoma Multiforme: Challenges and Opportunities for Drug Delivery

**DOI:** 10.3390/pharmaceutics15010059

**Published:** 2022-12-24

**Authors:** Harpinder K. Brar, Jiney Jose, Zimei Wu, Manisha Sharma

**Affiliations:** 1School of Pharmacy, Faculty of Medical and Health Sciences, The University of Auckland, Auckland 1023, New Zealand; 2Auckland Cancer Society Research Center, Faculty of Medical and Health Sciences, The University of Auckland, Auckland 1023, New Zealand

**Keywords:** blood–brain barrier, formulation strategies, glioblastoma multiforme, glioma, nanotechnology, tumor targeting, tyrosine kinase inhibitors

## Abstract

Glioblastoma multiforme (GBM) is an aggressive brain tumor with high mortality rates. Due to its invasiveness, heterogeneity, and incomplete resection, the treatment is very challenging. Targeted therapies such as tyrosine kinase inhibitors (TKIs) have great potential for GBM treatment, however, their efficacy is primarily limited by poor brain distribution due to the presence of the blood–brain barrier (BBB). This review focuses on the potential of TKIs in GBM therapy and provides an insight into the reasons behind unsuccessful clinical trials of TKIs in GBM despite the success in treating other cancer types. The main section is dedicated to the use of promising drug delivery strategies for targeted delivery to brain tumors. Use of brain targeted delivery strategies can help enhance the efficacy of TKIs in GBM. Among various drug delivery approaches used to bypass or cross BBB, utilizing nanocarriers is a promising strategy to augment the pharmacokinetic properties of TKIs and overcome their limitations. This is because of their advantages such as the ability to cross BBB, chemical stabilization of drug in circulation, passive or active targeting of tumor, modulation of drug release from the carrier, and the possibility to be delivered via non-invasive intranasal route.

## 1. Introduction

Gliomas stem from the glial or non-neuronal cells of the brain, including astrocytes, microglial, oligodendroglial, and ependymal cells. Previously, the World Health Organization (WHO) had classified gliomas into four grades based on mitotic activity, proliferation and degree of necrosis and glioblastoma multiforme (GBM) was categorized as a grade IV glioma [1]. However, this classic classification was purely histological and based on pathognomonic features. Therefore, the recent 2021 WHO classification for central nervous system (CNS) tumors requires the lack of isocitrate dehydrogenase 1 and 2 mutations (IDH-wildtype) as well as a lack of mutation in histone 3 (H3-wildtype) for the tumor to be diagnosed as GBM [2]. This classification presents molecular criteria that can be utilized for upgrading the diagnosis of histologically lower-grade, IDH-wildtype astrocytomas to glioblastoma, IDH-wildtype (WHO grade IV) [3]. These new classifications were mainly proposed to specify the prognosis of the diagnosis, because the patients with lack of these mutations have a poor overall prognosis than patients with the presence of the mutations [2].

GBM is one of the most aggressive and challenging to treat tumors. It displays considerable intratumoral heterogeneity and is characterized by genomic aberrations, high mitotic activity, microvascular proliferation, necrosis, resistance to apoptosis, and invasion into adjoining brain tissue [1]. GBM has poor prognosis with a median survival of around 12.6 months [4] as patients invariably relapse [5]. Recurrence is frequently within 2 cm of the original tumor margin, and recurrent GBM is usually not accessible to surgery and less sensitive to therapy than the initial tumor [5].

The current standard of treatment for GBM includes surgical resection, followed by radiotherapy and chemotherapy with temozolomide [4]. The diffuse and infiltrative nature of GBM makes complete surgical resection nearly impossible [1]. For chemotherapy, carmustine wafer implants received Food and Drug Administration (FDA) approval for recurrent glioblastoma in 1996 and for newly diagnosed glioblastoma in 2003 [6]. Temozolomide (for newly diagnosed GBM and as maintenance treatment) and bevacizumab (for recurrent GBM) were approved by the FDA in 2005 and 2009, respectively [6,7]. Since then there has been no marked improvement in the development of chemotherapeutic treatments for GBM that can increase the survival rates [4]. Other options have been explored, for example, tumor treating fields (TTFields), a non-invasive treatment approach, consisting of transcutaneous delivery of alternating electric fields with low intensity (1–3 V/cm) and intermediate frequency (100–300 kHz) [8]. TTFields was approved by the FDA for recurrent GBM in 2011 and for newly diagnosed GBM in 2015 [7]. However, it is expensive, has minimal survival benefits, and poor patient compliance [7,9,10]. Since the last decade, extensive research is ongoing to identify novel targets for the treatment of GBM in the drug discovery field. Meanwhile, formulation scientists are investigating novel drug delivery strategies to bypass or cross the BBB and to specifically target the drug to tumor.

In recent years, tyrosine kinase inhibitors (TKIs) have attracted attention because of their ability to target multiple pathways associated with GBM [11]. Tyrosine kinases (TKs) are essential regulators of cell signaling pathways and their activation leads to increased tumor cell growth, proliferation, initiation of anti-apoptotic pathways, and metastasis [11]. The receptor tyrosine kinase (RTK) signaling pathway has been identified as one of the critical pathways, abnormalities in which contribute to GBM initiation and progression in over 80% of patients [12,13]. Inhibiting TKs which are important regulators of cellular functions (proliferation, metabolism, migration, differentiation, and survival) and are required for cellular homeostasis, can lead to inhibition or slowing down of GBM cell proliferation and invasion into the surrounding brain microenvironment [14]. To date, more than 70 TKIs have been approved for the treatment of different cancers [15,16,17]. However, there is no approved TKI for treating GBM, and the clinical trials have been largely unsuccessful [12]. One of the main reasons behind such dismal outcomes is the poor BBB permeability of most TKIs [18]. In addition, the lack of tumor specificity may also be a reason behind limited efficacy [19]. Hence, various drug delivery approaches are being investigated to overcome the limitations of TKIs and increase their BBB permeability and tumor specificity.

This review briefly summarizes the role of tyrosine kinases in the pathophysiology of GBM and the potential of TKIs in the treatment of GBM. It also discusses the outcomes of clinical testing of TKIs in glioblastoma patients and the reasons for their limited efficacy. Furthermore, the review focuses on the drug delivery approaches that can be utilized to effectively deliver TKIs to the brain. These approaches include increasing BBB permeability, bypassing BBB by local administration or intranasal route, and using nanocarrier based drug delivery systems.

## 2. Pathophysiology of GBM and Role of TKs

### 2.1. Pathophysiology of GBM

Most GBM cases (90%) are primary and fast-growing tumors with no pre-existing lesion. Such tumors occur in older patients (mean age ~60) and are regarded as Grade IV tumors at the outset. However, secondary GBMs develop more commonly in younger patients (mean age <45) because of malignant progression from low-grade glioma [1]. Etiological risk factors linked to gliomas include, genetic factors, along with exposure to therapeutic ionizing radiation, pesticides, vinyl chloride, smoking, synthetic rubber manufacturing, and petroleum refining industries [20]. Genetic disorders including Ollier disease, Li-Fraumeni syndrome, and melanoma-neural system tumor syndrome also increase the risk of gliomas in children and adults [21]. GBM originates from glioblastoma stem cells (GSCs) [22]. They are highly proliferative, have strong tumorigenic abilities, and contribute resistance to radiotherapy by preferentially activating DNA-damage response pathways [20]. Glioblastoma can induce phenotypic modifications in normal cells. Normal cells collaborate with tumor cells and promote tumor proliferation, invasion of the brain, angiogenesis, immune suppression, and recruit normal cells to protect tumor cells from the effect of chemotherapy or radiotherapy. In this takeover of the brain by glioblastoma, several modes of communication are involved, including but not limited to cell-secreted soluble factors, gap junctions, extracellular vesicles (microvesicles and exosomes), and nanotubes. The cell-secreted soluble factors include transforming growth factor-β (TGFβ), Notch, interleukin-6 (IL-6), platelet derived growth factor (PDGF), epidermal growth factor (EGF), VEGF, and stromal cell-derived factor 1 (SDF1) [23].

Some of the important pathways other than TKs involved in development of GBM are discussed as follows.

#### 2.1.1. p53 and PTEN Pathway

Tumor protein p53 is a tumor suppressor protein that induces DNA repair or apoptosis in case of irreparable DNA damage. A strong correlation exists between the presence of mutant p53 and the transition from low-grade astrocytoma to high-grade glioblastoma [24]. Further, nuclear p53 induces apoptosis and limits tumor cell expansion, and it has been reported that nuclear localization is associated with long-term survival rates [25]. It has also been observed that primary tumors with p53 mutations had concomitant PTEN mutations or deletions in 60% of human primary GBM samples [26]. Mutations in PTEN, a phosphatase tumor suppressor gene, are seen in 5–40% of glioblastomas. PTEN assists homeostasis by preventing cell cycle entry and therefore, maintaining the population of neural stem cells. Null mutants of PTEN are more sensitive to growth factors and more susceptible to proliferation than wild-type neural stem cells [4].

#### 2.1.2. Isocitrate Dehydrogenase (IDH) Pathway

Mutations in IDH-1 are critical in the transition of low-grade gliomas to secondary GBM. These mutations are rare (5%) in primary GBM; however, they are observed in 83% of all secondary GBMs [27]. They are also believed to be one of the initial events in the development of low-grade gliomas before any mutation that may take place in p53 gene [28]. In addition, IDH2 mutations are also more frequent in secondary glioblastomas than in primary glioblastomas [29]. Both IDH1 and IDH2 mutant GBM are associated with better survival rates than IDH wild-type GBM [29,30].

#### 2.1.3. Retinoblastoma (RB) Pathway

The RB pathways comprise of CDKN, D-type cyclins, cyclin-dependent protein kinases, E2F-family of transcription factors, and retinoblastoma tumor suppressor gene (RB1). The pathway plays a crucial role in regulating cell proliferation and is often altered in various cancer types [31]. Alterations in RB pathway have been reported to be observed in 78% of glioblastomas [32,33].

#### 2.1.4. Histone H3 Pathway

H3K27M mutation, a methionine substitution for lysine at residue 27 of histone H3, has also been reported in glioblastoma. Although this mutation is associated with malignant pediatric diffuse midline glioma; however, it is reported that it may possess typical anatomical preferences in glioblastoma and cerebellar location is one such location where this mutation has been consistently reported [34].

#### 2.1.5. Interleukins (ILs) Pathway

The microenvironment of glioblastomas consists of chemokines, pro-inflammatory cytokines, and growth factors. It has been observed that patient samples and GBM cell lines have significant up-regulation of interleukins IL-1β, IL-6 and IL-8 and some of them also have prognostic potential. Amplification of IL-6 gene directly correlates with glioblastoma aggressiveness leading to decreased patient survival. These ILs activate janus kinase (JAK), p38 mitogen activated protein kinase (MAPK), and c-Jun N-terminal kinase (JNK) signaling pathways, resulting in proliferation, invasiveness, and angiogenesis in GBM [35].

### 2.2. Role of TKs in Pathophysiology of GBM

Tyrosine kinases (TKs) can be categorized as receptor TKs (RTKs) and non-receptor TKs (nRTK), the latter is also known as cytoplasmic TKs. RTKs are based on cell surface receptors and include receptors like epidermal growth factor receptor (EGFR), vascular endothelial growth factor receptor (VEGFR), platelet-derived growth factor receptor (PDGFR), hepatocyte growth factor receptor (HGFR), fibroblast growth factor receptor (FGFR), and insulin-like growth factor 1 receptor (IGF-1R); whereas nRTKs are cytoplasmic proteins and include focal adhesion kinase (FAK), c-SRC, and JAK. The specific molecular mechanism depends on the type of TK involved [21].

Figure 1 depicts the structure of TK receptors. Ligand binding to RTK (Figure 2) results in activation of the receptor, which further leads to receptor dimerization and autophosphorylation of TK domain. Two main downstream signaling pathways activated by this event include Ras/MAPK/extracellular-signal-regulated protein kinase (ERK) and Ras/phosphatidylionositol-3-kinase (PI3K)/Akt and these are implicated in proliferation, invasiveness, angiogenesis, and survival [13].

EGFR: EGFR was one of the first proto-oncogenes that was found to be potentially associated with GBM pathogenesis [37]. PI3K and MAPK pathways are activated by EGFR, resulting in cell proliferation and angiogenesis [21]. EGFR plays a role in the pathogenesis of GBM and also resistance to treatment [13]. Around 60% of GBM patients have some type of genomic alteration affecting EGFR pathway [19]. The majority of the mutants have a deletion in the N-terminal ligand-binding region between amino acids 6 and 273 named EGFRvIII (mutated EGFR that can induce transformations of surrounding cells to GBM-like phenotypes), which can result in ligand-independent activation of EGFR [38].

VEGFR: GBM is a highly vascularized tumor and anti-angiogenic therapies have been widely investigated for its treatment [39]. VEGF is implicated in the process of angiogenesis [40]. It is activated in hypoxic conditions by translocation of hypoxia-inducible transcription factors (HIF1α and HIF1β) to the nucleus. VEGF activation results in increased angiogenesis to counteract hypoxia. GBM tumors are often hypoxic and have enhanced expression of VEGF, which leads to irregular vasculature and up-regulation of VEGFR in GBM [13].

PDGFR: PDGF is involved in cell cycle regulation (cell cycle initiation, DNA synthesis, and mitosis), cell migration, and chemotaxis. Overexpression of PDGF has also been shown to induce tumors in experimental animals [41]. Lane et al. reported that inhibition of PDGFR can be used for treatment of GBM by initiating neuronal differentiation in tumor cells, which subsequently reduces tumor development [42].

HGFR: HGFR or mesenchymal-epithelial transition factor (c-Met) augments malignancy by inducing cell proliferation, survival, migration, and invasion, promoting tumor angiogenesis, and supporting a stem cell phenotype [43]. Overexpression of HGFR is frequently observed in 29–88% of glioblastomas [44]. Its expression in human gliomas is associated with higher grade and worse clinical outcomes [45].

FGFR: Basic FGFR is a potent mitogen and angiogenic peptide and reported to be an autocrine regulator of glioma cell growth [46]. FGFR has also been reported to stimulate the growth of cultured GBM cell lines and inhibition of FGFR by RNA interference or monoclonal antibody limited proliferation of GBM cells [47].

IGF-1R: IGF-1R receptor on activation promotes glioma cell proliferation and migration and may also trigger low-grade gliomas to progress to GBM [48]. IGF-1R overexpression in GBM was linked to reduced survival and reduced responsiveness to the approved drug temozolomide [49].

nRTKs: The activity of intracellular tyrosine kinases (nRTKs), which are important regulators of signal transduction from surface receptors, is also elevated in malignant cells. These kinases comprise of proteins that play a role in signaling cascades like the mammalian target of rapamycin (mTOR), PI3K/AKT, MAPK/ERK, JNK, SRC, and JAK and signal transducer and activator of transcription (STAT) [50]. SRC, a downstream signaling intermediate of many RTKs, initiates phosphorylation of several substrates and promotes regulation of pathways associated with cell survival, proliferation, adhesion, motility, and angiogenesis [51]. Mutation in the c-SRC pathway results in detachment of tumor cells by interference with integrin, which promotes metastasis [21]. In addition, its activity encourages and sustains inflammation and metabolic reprogramming in the tumor microenvironment, thus supporting tumor growth [51]. It was also reported that glioblastoma cells were sensitive to JAK2 inhibition [50].

## 3. Preclinical GBM Models

For the clinical success of treatments in GBM patients, it is imperative to have an authentic tumor model that can capture the genetic and phenotypic properties of human glioblastoma. Currently available models are not perfect, and reproducing properties of human tumors, especially the glioblastoma microenvironmental communication, is very challenging [23,52]. The models used for brain tumor studies include syngeneic models, genetically engineered models (GEMs), and xenografts (cell line-based and patient derived) and have been summarized by Akter et al. [52]. Alphandéry E. has also summarized the small (mice, rats) and large (dogs) animal models used for GBM studies [53].

One way of improving the tumor model is that cells from patient tumors can be isolated and maintained as neurospheres or organoids in serum-free media to retain the genetic heterogeneity and the GSCs. Patient-derived xenograft models consist of implanting and passaging portions of patient tumors in immune-compromised mice. In syngeneic mouse models, tumors are first induced by chemicals or viruses and are established as cell lines to be transplanted back into the mouse brain. Further, spontaneous brain tumors can also be induced with known driver mutations in GEMs [23]. However, these models have a few limitations, such as, cell lines, neurospheroid cultures, and patient-derived xenograft models suffer from genetic instability. Glioma-derived cells also show a different genomic methylation pattern and transcriptome in culture and in vivo. GEMs demonstrate only a few driver mutations and have less neoantigens. Hence, it is more suitable to use more than one type of mouse tumor model for testing [23,52].

## 4. Tyrosine Kinase Inhibitors (TKIs)

TKIs inhibit RTKs or cytosolic TKs. Inhibition of TKs can be accomplished through mechanisms including direct competition for ATP binding to TK, allosteric inhibition of TK, inhibition of ligand binding to RTKs, inhibition of interaction of TK with other proteins, or destabilization of TK [54]. Blockage of cell signaling results in inhibition of cell growth, proliferation, differentiation, and angiogenesis. Imatinib, which targets the BCR-Abl kinase, was the first TKI approved in 2001 and is successfully used for the treatment of chronic myeloid leukemia [55]. After imatinib, several other TKIs were approved for treatment of different cancers. TKIs have shown promising results in preclinical trials for treatment of GBM and brain metastasis. For instance, Imatinib enhanced the radiosensitivity of U87 human glioblastoma cells and increased reduction in tumor growth induced by fractionated radiotherapy [56]. Targeting RTK tunica interna endothelial cell kinase 2 (TIE-2) by a highly potent inhibitor BAY-826 improved tumor control in in vivo mouse model of glioma [57]. Additionally, altiratinib, a Met/TIE-2/VEGFR2 inhibitor, in combination with bevacizumab has been reported to significantly decrease tumor volume, invasiveness, microvessel density, expression of mesenchymal marker, and TIE-2 expressing monocyte infiltration in glioblastoma xenograft mouse models as compared to bevacizumab monotherapy [58].

Various EGFR inhibitors have been investigated in relation to GBM. Osimertinib, an EGFR inhibitor, has shown BBB permeability in a mouse model [59]. It also significantly inhibited the growth of six different GBM cell lines and significantly prolonged survival of GBM-bearing mice [60]. Lazertinib, another EGFR inhibitor, effectively inhibited intracranial tumor growth in an EGFR-mutant mouse model of brain metastasis and its efficacy was more than Osimertinib [61]. Further, gefitinib (an EGFR inhibitor) radio sensitized GBM cell lines in vitro [62]. Afatinib (an EGFR inhibitor) in combination with temozolomide significantly inhibited proliferation and invasion of U87 and U251 cells and delayed tumor growth and progression in preclinical mouse models [63].

Apart from these small molecule TKIs, various natural products have also been found to inhibit TKs. Natural products play a crucial role in discovering and developing new anticancer drugs. Yin et al. have reviewed various natural products and derivatives that have been reported to inhibit different TKs [64].

## 5. Clinical Trials of TKIs in Treatment of GBM and Disappointing Outcomes

Due to the effectiveness of TKIs in various tumors and their potential in GBM, clinical studies were conducted to test their utility in GBM. Clinical outcomes of TKIs in adult glioblastoma patients are not highly encouraging. Clinical trial results have been disappointing and disadvantaged mainly by broad inclusion criteria, poor pharmacokinetics of TKIs, and resistance to TKIs [65]. For inclusion criteria, selecting patients according to molecular signature of their tumor can increase the possibility of response [66]. In majority of the clinical trials where TKIs are used, inclusion criteria does not take into account the expression of the RTK targeted by the drug. Moreover, most clinical trials with TKIs are conducted in patients with recurrent GBM in which carrying out target expression studies is harder. Recurrent GBM are less expected to undergo surgery and their genetic landscape is different from that of the original tumor [5]. Resistance to TKIs can be due to mechanisms including activation of alternative receptors or signaling pathways and cell adaptation to a new environment [67]. TKIs selectively inhibit one or multiple RTKs, but GBM cells may compensate by activating several TKs [12].

In addition, most TKIs suffers from poor BBB permeability, which limits their efficacy in brain tumors [68]. Portnow et al. determined neuropharmacokinetics of bafetinib, which targets BCR-Abl kinase, using intracerebral microdialysis in adult patients with recurrent high-grade gliomas and reported that it was not able to sufficiently cross either intact or disrupted BBB after systemic administration [69]. Further, Mehta et al. [70] carried out a phase 0 trial to measure the tumor pharmacokinetics and pharmacodynamics of ceritinib, an inhibitor of anaplastic lymphoma kinase (ALK), in patients with recurrent GBM and brain metastasis. They reported that ceritinib is largely bound to plasma proteins and tumor tissues and unbound drug concentrations were insufficient for target modulation in the patients. The observations indicated that ceritinib has minimal penetration to tumors and hence do not have the potential to be pursued as an anticancer drug in these tumors. Moreover, most of the anti-cancer drugs are substrates for ATP-binding cassette (ABC) efflux transporters, contributing to the low drug accumulation in the brain [5]. Brain uptake of TKIs including gefitinib, regorafenib, and tivozanib is reported to be restricted by ABC transporters [71,72,73]. Even if the TKI is able to cross the BBB, evidence indicates that TKIs do not reach sufficient intra-tumoral therapeutic concentrations [5].

TKIs are also reported to be substrates for cytochrome P450 enzymes. Their metabolites are more hydrophilic and may have less kinase selectivity and more off-target interactions than the parent drug [74,75]. Therefore, understanding the pharmacokinetics and pharmacodynamics of TKIs in the brain is also essential for their future applications in GBM [14].

Further, different molecular pathways are simultaneously activated in GBM, leading to tumor heterogeneity [76]. Brain tumors work as an ecosystem with interconnected networks, rendering treatment with single-agent therapeutics largely inefficacious. Thus, a combination of drugs targeting different vital pathways can be more efficacious, considering the past failures of many single agents [77]. A study by Stommel et al. suggested weak response with RTK-inhibitor monotherapy and more promising outcomes with a combination of drugs acting against different activated RTKs or a single drug that can act against multiple activated RTKs [78]. Brown et al. also noticed a trend towards improved survival and response rates in patients with recurrent GBM when gefitinib (EGFR inhibition) was added to cediranib (VEGF inhibition) [79]. Further, it has been observed that RTK systems can co-modulate different and overlapping downstream signaling pathways that are leading to cancer. For instance, c-MET and EGFR are associated with the malignant progression of GBM. EGFRvIII variant of EGFR results in enhanced tumor growth and reduces the tumor growth response to HGF: c-MET pathway inhibitor treatment. On the other hand, activation of c-MET pathway reduces the tumor growth response to inhibitors of EGFR pathway. The use of combination of c-MET and EGFR inhibitors to target these two pathways has been reported to provide considerable anti-tumor activity in glioblastoma models [80].

Phase II and III clinical trials of various TKIs with more than 30 participants are included in Table 1.

## 6. Pediatric High-Grade Gliomas (pHGGS)

Pediatric high-grade gliomas (pHGGs) constitute one-tenth of pediatric brain tumors [97]. They have a poor prognosis and are clinically and biologically different from the disease in adults [98]. The most frequent mutations observed in pHGGs are gene amplification of RTKs including PDGFRA, EGFR, KIT, IGF1R, and MET, homozygous inactivation of p53 and histone 3.3 (H3F3A), and deletion of CDKN2A, TP53, and ADAM3A [99]. No apparent beneficial effect of the addition of chemotherapeutic drugs such as temozolomide to radiotherapy after surgical resection over radiotherapy alone has been observed in pediatric population [100]. Hence, novel targeted therapies are needed to treat pediatric brain tumors. TKIs have been postulated to have therapeutic efficacy in pHGGs, as 62% of clinical samples had aberrations in RTK signaling pathway [65,101]. However, they have shown limited efficacy in clinical trials. Sun et al. have discussed the clinical trials of TKIs in pHGGs and some of the reasons behind limited efficacy [65]. Drug delivery approaches for effectively delivering TKIs to the brain can also prove to be beneficial for treating pHGGs, for which effective treatment options are lacking.

## 7. Strategies to Achieve Targeted Delivery of TKIs

Drug delivery strategies have played a crucial role in the conversion of promising therapeutics into successful treatments. Delivery strategies have evolved over time in accordance with changing drug delivery needs and evolvement of the therapeutic landscape [102]. Effective drug delivery during chemotherapy is mainly challenged by factors such as hypoxic tumor environment, phenotypic and genotypic heterogeneity, presence of GSCs, aberrant signaling pathways, and most prominently by the presence of the BBB [103]. BBB is composed of compactly sealed brain capillary endothelial cells with tight junctions and provides optimum conditions for brain functions by supplying brain with nutrients, preventing entry of harmful substances through circulation, and restricting movement of fluids and ions [103,104]. BBB limits brain penetration of majority (> 98%) of small molecules, and only the molecules that are not substrates for active efflux transporters and fit Lipinski’s rule of five are likely to penetrate BBB by passive diffusion [105].

Furthermore, BBB is modified in malignant gliomas to form blood-brain-tumor barrier (BBTB), which is characterized by non-uniform delivery and active efflux of molecules [106]. BBTB still maintains the properties of BBB and the new blood vessels in brain tumor are less leaky than angiogenic vessels in other type of tumors [107]. Sarkaria et al. reported that BBB is not uniformly disrupted in all GBM patients and clinical evidence shows that there is a clinically significant tumor burden with an intact BBB in GBM. Therefore, drugs having poor BBB permeability will not provide therapeutically effective drug exposure to this part of tumor cells [108].

This makes drug delivery to tumor sites challenging, necessitating innovative approaches to overcome the barriers. Moreover, novel drug delivery strategies are also required to target and treat the parenchyma-infiltrating GBM cells which remain after the surgical resection, invade the surrounding brain parenchyma, and are resistant to chemotherapy and radiotherapy [109]. Several approaches are being investigated to increase transport of drugs and overcome the BBB in treatment of brain tumors. Some of these approaches include increasing the permeability of BBB, bypassing the BBB, using nanotechnology to cross the BBB, inhibition of efflux transporters, and chemical modification of drug molecules [106,110].

### 7.1. Altering BBB Permeability

BBB can be disrupted chemically or physically to increase permeability by techniques such as the alteration of tight junctions through the administration of bradykinin analogues, osmotic disruption by mannitol, and focused ultrasound techniques [106].

#### 7.1.1. Chemical Disruption

Minoxidil sulfate is a selective activator of ATP-sensitive potassium channels, which are overexpressed on BBTB but are rare in normal brain capillaries and can be targeted for modulation of BBTB permeability. A study conducted in rats demonstrated improved selective BBTB permeability and delivery to brain tumor of carboplatin in presence of minoxidil sulfate with increase in survival rate of up to 38% as compared carboplatin alone [111]. Similarly, calcium dependent potassium channels are also overexpressed on tumor cells as compared to normal cells. NS-1619, an agonist of these channels, was able to increase BBTB permeability and carboplatin delivery to brain tumor resulting in increased survival in rat syngeneic and xenograft brain tumor models [112]. Possible ways reported for achieving better BBB penetration of TKIs in GBM includes the use of methamphetamine, a FDA approved drug for attention deficit disorders, which has shown potential to open the BBB in rodents and use of selegiline, a monoamine oxidase inhibitor, as it is metabolized to methamphetamine [113].

#### 7.1.2. Focused Ultrasound

It has been established that treatment with non-thermal burst-mode ultrasound in the presence of microbubbles can induce local and reversible opening of BBB by causing disruption of tight junctions in CNS capillaries. Microbubble-facilitated focused ultrasound was able to open BBB and allow penetration of therapeutic substances across BBB in preclinical settings [114]. Moreover, the use of pulsed ultrasound along with systemic microbubble injection before treatment with carboplatin was reported to be feasible in patients with glioblastoma [115]. However, this technique was not found to be of use in the case of erlotinib. Disruption of BBB with focused ultrasound did not improve the delivery of erlotinib to the brain in rats. However, it was improved by inhibiting efflux transporters P-glycoprotein (P-gp) and breast cancer resistant protein (BCRP). Therefore, delivery of erlotinib to brain is mainly dependent on ABC transporters and not on BBB integrity and it is therefore important to know the physical and biological mechanism behind the limited brain distribution of a particular TKI [116].

#### 7.1.3. Inhibition of Efflux Transporters

Inhibiting efflux transporters is another way to increase BBB penetration of drugs. P-gp is highly expressed on BBB and glioblastoma cells and limits therapeutic efficacy of various chemotherapeutics by inducing their brain-to-blood efflux. Thiosemicarbazone compounds were reported to be effective inhibitors of P-gp efflux transporter in BBB and glioblastoma cells. In nanomolar concentrations, these compounds significantly increased retention of P-gp substrate drugs in the BBB [117]. Several TKIs are also reported to have interaction with ABC transporters such as P-gp, BCRP, and multidrug resistance protein 1 (MRP1) [118]. As discussed above, erlotinib delivery to brain was enhanced by inhibition of P-gp and BCRP transporters [116]. However, many TKIs interact with ABC transporters both as substrates and inhibitors. They normally tend to act as substrates at lower concentrations and inhibit the transporters at higher but pharmacologically relevant concentrations [119].

#### 7.1.4. Chemical Modification of Molecules

Drug modification can be done by chemical alteration of its structure (alteration of a functional group, masking undesirable groups, or PEGylation) and its conjugation to targeting ligands or known moieties [102]. Physicochemical properties of a drug can be modified such that the resulting molecule has increased BBB permeability [110]. For instance, addition of a succinate group to C10 of paclitaxel resulted in reduced interaction with multidrug resistant type 1 (MDR1) and increased BBB permeability [120]. However, increased BBB permeability because of enhanced lipophilicity of drugs does not necessarily mean increase in their efficacy, because it can increase non-specific binding of drug to brain tissue and reduce its availability for therapeutic target. Therefore, encapsulation of therapeutic agents in nanoparticles have been utilized to overcome this challenge of reduced efficacy in attempt to increase BBB permeability [110]. Further, poly(ethylene glycol) (PEG) coating is an effective approach to increase the circulation half-life of particles and their retention at tumor sites [121,122]. In recent years, a class of dyes called heptamethine cyanine dyes (HMCDs) were utilized to chemically attach various FDA-approved drugs to improve the BBB penetration and tumor tissue specificity [123,124]. They are being investigated as a potential DDS to deliver non-selective chemotherapeutics to tumor and overcome issues like BBB permeability and tumor specificity [106]. The use of HMCDs as drug-conjugate systems (including TKIs) has been reviewed previously by various groups and hence will not be covered in this review [106,123,124].

#### 7.1.5. Intra-Arterial Drug Delivery

In intra-arterial drug delivery, drugs are directly administered into an artery close to the tumor. A hyperosmolar agent such as mannitol, can also be administered along with drug to open the BBB locally. However, its utility in primary brain tumors is limited by toxicity and low drug efficacy [125].

### 7.2. Bypassing BBB via Alternate Routes

Direct intracerebral/intratumoral injection, implantable polymeric systems, convection enhanced delivery (CED), and intranasal (nose-to-brain) delivery are the main approaches to bypass BBB [103,126]. CED and intranasal delivery can help to overcome some limitations of the traditional methods such as implantable polymeric systems and direct injection.

#### 7.2.1. Direct Injection

Direct injection of the agent can be given in the tumor resection cavity or nearby brain parenchyma and can be achieved through needle-based injections or catheter implants connected to a reservoir [110]. Intracerebral or intratumoral injection avoids the BBB, decreases systemic side-effects, and increases the concentration of drug at tumor site for effective therapy. However, penetration of the drug into the tumor parenchyma is not certain by this method because of limited drug diffusion and there is also an increased risk of local side effects such as infections and intracranial hemorrhage [127]. Additionally, when catheters are used, there is a problem of blockage of catheter by tissue debris [126].

#### 7.2.2. Implantable Polymeric Systems

Alternative to direct injection is the implantation of drug-loaded polymeric systems like wafers, gels, and microchips in the post-operative cavity. This allows for gradual release of drugs from the polymers, less systemic side-effects, extended duration of treatment, and addition of multiple chemotherapeutic agents [126]. Gliadel wafers are biodegradable polymers impregnated with carmustine that are implanted in tumor bed after its resection and provide controlled release of drugs for up to three weeks [128]. They are approved for treatment of newly diagnosed and recurrent GBM, but they are associated with incidence of adverse events such as cerebral edema, intracranial hypertension, healing abnormalities, cerebrospinal fluid leaks, intracranial infection, and seizures [129].

Hydrogels are also being explored for local treatment of GBM. Hydrogels are 3-dimensional polymeric hydrophilic networks in an aqueous medium that can encapsulate different drugs and provide controlled release. They can prove to be more beneficial than the Gliadel wafers because of their similarity with brain tissue in terms of mechanical properties and softness [109]. Bastiancich et al. have provided in-depth discussion on anticancer drug loaded hydrogels for local treatment of GBM [127].

Innovative sprayable drug delivery systems are also under investigation for localized drug delivery to GBM [130]. McCrorie et al. developed a sprayable bio-adhesive hydrogel to improve the adaptability to the GBM resection cavity [131].

#### 7.2.3. Convection Enhanced Delivery (CED)

It involves pressure-driven bulk flow of infusate from a pump via a pre-set catheter for local delivery of agents [126]. It is a promising method for treatment of diseases that are not responsive to systemic therapy and can also avoid systemic side effects [132]. A phase I study of CED of ^124^I-8H9 in pediatric patients with diffuse intrinsic pontine glioma (DIPG) having received radiation therapy showed that CED is a rational and safe approach to achieve high dose in the lesions with insignificant systemic toxicity [133]. Thus, Tosi et al. labeled different modifications of dafatinib through nanofiber-Zirconium-89 system or with Fluorine-18 to study effect of these modifications on in vivo glioma efficacy and pharmacokinetic behaviour in brain parenchyma via positron emission tomography (PET). Relatively similar bioactivity was observed in animal-model and patient-derived cell lines of DIPG. However, significant individual variability was observed in CED drug clearance in naive mice [132]. There are numerous challenges in front of CED including catheter backflow, more understanding of pharmacokinetics, and optimization of therapeutics. Therapeutic effectiveness of this approach is not clear. It was able to achieve regional disease control, but failed to cover other distant areas and lead to spread of disease [134]. D’Amico et al. have reviewed the history and principles of CED, advancements in the procedure, outcomes of important clinical trials of CED in GBM, and potential future of this technique [135].

#### 7.2.4. Intranasal Delivery

This non-invasive strategy has gained interest in the recent years for brain drug delivery. Nose-to-brain delivery is an effective non-invasive drug delivery approach to bypass the BBB and reduce the systemic side-effects of drugs, thereby providing optimal treatment for GBM patients [136]. Transport of drugs from nasal mucosa to brain mainly takes place through olfactory and trigeminal nerves by intracellular/extracellular pathways or through perivascular channels of lamina propria [137]. Nanoparticles are being explored for nose-to-brain delivery of drugs because of their advantages such as protection of the drug from biological and chemical degradation, prevention from ABC transporter mediated efflux, increase in residence time by use of mucoadhesive formulations, and opening of mucosal epithelium tight junctions because of use of surfactants [137]. Bruinsmann et al. have summarized the preclinical trials of drugs and nanoparticles being tested for nose-to-brain delivery in GBM and clinical trials on use of intranasal perillyl alcohol for GBM. Perillyl alcohol is the sole agent administered intranasally for GBM treatment to reach clinical trials. However, it is given via an inhalation protocol, which might not completely involve mechanism for nose-to-brain delivery [136].

### 7.3. BBB Crossing via Nanotechnology

Nanocarriers are evolving as a dominant drug delivery platform for treatment of brain tumors [107]. Nanocarriers assist drug delivery to brain tumors via various strategies that include chemical stabilization of the chemotherapeutic agent in systemic circulation, passive targeting, inhibition of efflux transporters, and active targeting of carriers and receptors overexpressed at BBB [138]. Various delivery systems such as polymeric nanoparticles, magnetic nanoparticles, dendrimers, lipid based systems including liposomes, nanostructured lipid carriers, and solid lipid nanoparticles can be utilized for effective delivery of drugs to brain [139]. Nanocarriers that can adequately retain the drug during circulation and release it on accumulation at tumor site can greatly improve the therapeutic index of the loaded drugs [140].

Leaky blood vessels of the tumor tissue enable passive targeting of non-targeted nanoparticles via enhanced permeability and retention (EPR) effect. However, non-targeted nanoparticles can also accumulate at normal tissues and release the drug within healthy cells, causing side-effects. This limitation can be overcome by active targeting, which enhance the specificity of nanocarriers towards cancer cells and reduce the risk to healthy cells. Active targeting can be accomplished via ligands such as aptamers, antibodies, peptides, and proteins [141]. Endogenous transport processes such as receptor-mediated transcytosis (RMT), carrier-mediated transcytosis, adsorptive-mediated transcytosis, and cell-mediated transcytosis can be utilized to deliver the nanoparticles across the BBB. Among these RMT is primarily exploited by utilizing nanoparticles modified with various ligands that can bind specific receptors on the BBB [141,142]. These receptors include transferrin receptor, low-density lipoprotein receptor, low-density lipoprotein receptor-related protein, folate receptor, lactoferrin receptor, insulin receptor, diphtheria toxin receptor, neonatal Fc receptor, nicotinic acetylcholine receptor, nucleolin receptor, scavenger receptor class B type, and leptin receptor [141]. Drug release can also be modulated by additional approaches such as pH dependent release to achieve better effects [74].

Moreover, nanoparticles can be used to target the invasive GBM cells near tumor margins that are left behind after surgical resection and the GSCs. Fibroblast growth factor inducible 14 (Fn14) is a member of TNF receptor superfamily and Fn14-targeted nanoparticles are being explored in invasive glioma biology. Nestin and Prominin are putative GSC cell surface markers and nanoparticles targeting GSCs for GBM treatment are also being investigated [142].

Ganipineni et al. have discussed the recent preclinical trials of nanomedicine for active targeting of GBM and the clinical trials of nanomedicine for GBM treatment [143]. Nanocarrier vehicles investigated for delivery of TKIs in high-grade brain tumors are summarized in Table 2 and briefly discussed subsequently. Figure 3 describes various drug delivery approaches that can be utilized for targeted delivery to brain.

#### 7.3.1. Liposomes

Liposomes are composed of concentric single or multiple lipid bilayers with an aqueous core. Inherent qualities of liposomes, such as their morphological similarity to cellular membranes, ability to incorporate both hydrophilic and lipophilic drugs, biocompatibility, and non-immunogenicity make them the most-successful drug delivery system [144]. Lakkadwala et al. constructed surface modified liposomes for efficient delivery of doxorubicin and erlotinib (targets EGFR RTK). They utilized transferrin to target transferrin receptors overexpressed on glioblastoma (U87 cells) and brain endothelial (bEnd.3) cells and a cell-penetrating peptide (PFVYLI) to improve intracellular uptake of carriers [15,145]. Further, Dasatinib is an inhibitor of Src family Kinases, BCR-ABL, PDGFR, and c-Kit [146]. Benezra et al. formulated micellar and liposomal nanoformulations for a novel fluorinated dasatinib derivative (^18^F-SKI249380) and employed microPET to assess drug delivery and uptake [147].

#### 7.3.2. Polymeric Nanoparticles

Polymeric nanoparticles can be composed of natural polymer nanomaterials like chitosan, cellulose, alginate, gelatin, and hyaluronic acid or synthetic polymers like polyvinyl alcohol, polycaprolactone, polyethylenimine, poly-lactide-co-glycolic acid (PLGA), and polylactic acid. They are widely used for targeted delivery to various cancers [148]. Khan et al. functionalised optimized imatinib mesylate (competitive inhibitor of ATP binding to Abl kinase, c-Kit, and PDGFR) loaded PLGA nanoparticles with Pluronic^®^ P84, a P-gp inhibitor, to increase therapeutic concentration of drug in the tumor [149,150]. Another group of researchers associated anti-PDGFRβ aptamer (Gint4.T) with polymeric nanoparticles composed of PLGA-block-PEG copolymer for delivery of dactolisib, a potent PI3K-mTOR inhibitor [151].

#### 7.3.3. Polymeric Micelles

Amphiphilic block copolymers can self-associate in aqueous solution to form micelles [152]. Such polymeric micelles are widely used as drug carriers because of their properties such as better thermodynamic stability, narrow size distribution, core–shell structure, suitability as carriers for hydrophobic drugs, and easy surface modification or stimuli sensitization [152,153]. Wei et al. reported that apolipoprotein E peptide has high affinity for LDLRs and mediate BBB penetration and protein delivery for orthotopic GBM. Hence, they investigated LDLR-specific micelles composed of PEG-*b*-poly(ε-caprolactone-co-dithiolane tri-methylene carbonate)-mefenamate copolymer for loading of sorafenib, a multi-kinase inhibitor of RAF-MEK-ERK pathway, VEGFR, and PDGFR [154]. Nehoff et al. observed that the combination of pan-kinase inhibitors crizotinib and dasatinib was most potent among the 12 TKIs tested in established and primary human GBM cell lines [155]. Therefore, the same group formulated poly(styrene-co-maleic acid) micelles for encapsulation of crizotinib and dasatinib to achieve better targeted therapy and reduced systemic toxicity in GBM [156].

#### 7.3.4. Albumin Nanoparticles

Albumin, the most abundant protein in plasma, has been an appealing nanoscale drug carrier with a great safety profile. Its accumulation in solid tumors via the EPR effect is the rationale behind developing albumin-based drug delivery systems for tumor targeting. Abraxane (albumin bound paclitaxel) has been clinically approved for treating metastatic breast cancer [157]. Recently, Yang et al. developed human serum albumin based nanoparticles for co-delivery of ibrutinib (inhibitor of Bruton’s TK) and hydroxychloroquine, and proved that the nanoparticles accumulated at the tumor site after intravenous injection and prolonged the survival in animal model of glioma [158].

#### 7.3.5. Inorganic Nanocarriers

Inorganic nanocarriers including silica, gold, carbon nanotubes (CNTs), and graphene have also been utilized as drug delivery vehicles, because of their versatile physicochemical properties such as low cytotoxicity, easy availability and functionalization, and accumulation in tumor cells without recognition by P-gp [159]. For example, a group of researchers employed ultra-small fluorescent core–shell silica nanoparticles (Cornell prime dots or C’ dots) comprising of a rigid silica core encapsulating a fluorescent dye (Cy5), surface covalently modified with PEG chains, and integrin-targeting peptide and dasatinib conjugated to the surface [160]. Additionally, Moore et al. formulated CNT with multiple polymer coatings to improve the release kinetics and therapeutic efficacy of dasatinib. However, there are concerns related to toxicity of CNTs because of similarity of fibrillar CNT to asbestos [161]. Yet, studies have reported that toxicity is dependent on shape, size, dose, route of administration, and surface functionalization of nanoparticles and CNTs can be safely administered and metabolized [162].

#### 7.3.6. Other Nanocarriers

Lipid nanocapsules have many advantages as drug delivery vehicles like fabrication by a phase-inversion process utilizing excipients that are generally recognized as safe (GRAS), devoid of organic solvents, high drug loading, high stability, and the opportunity of easy scale-up of their production [163,164]. Clavreul et al. formulated lipid nanocapsules loaded with sorafenib to overcome the limitations of free drug [164]. Further, Yu et al. utilized PEGylated bilirubin nanoparticles modified with D-form T7 (D-T7) peptide as a DDS for cediranib and paclitaxel. Cediranib inhibits the tyrosine kinase activity of c-KIT and VEGFR1-3. D-T7 was used for targeting glioma as it has high affinity for transferrin receptor (TfR), overexpressed on endothelial and brain capillary glioma cells. Further, PEGylated bilirubin nanoparticles have been utilized as drug carriers for treatment of various inflammatory disorders and they have also been reported to be fairly water-dispersible, circulating in blood circulation for much longer times, and able to be selected as dual-responsive (reactive oxygen species and light) DDS [165].

**Table 2 pharmaceutics-15-00059-t002:** Nanocarriers utilized for delivery of TKIs to brain tumors in animal models.

Formulation	Drug/Targeting Ligand	Particle size	Cell line/Model Used	Outcomes/References
Actively targeted nanoparticles				
Ultra-small fluorescent core–shell silica NPs	Dasatinib; cyclic-arginine-glycine-aspartic acid peptide	6–7 nm	TS543 neurosphere cultures Genetically engineered mouse model of glioblastoma	Effective inhibition of PDGFR signaling in vitro Enhanced in vivo accumulation, diffusion, and retention [160]
Polymeric NPs based on PLGA-b-PEG-COOH	Dactolisib; Gint4.T aptamer	-	U87MG GBM cells Nude mice bearing intracranial U87MG tumor xenografts	Higher uptake of NPs in vitro EC50 value of 38 pM on loading with the drug Specific GBM tumor uptake in vivo after IV administration [151]
PEG-*b*-poly(ε-caprolactone-co-dithiolane trimethylene carbonate)-mefenamate micelles	Sorafenib; Apo-lipoprotein E peptide	24 nm	U-87 MG cells U-87 MG-Luc tumor-bearing mice	Good in vivo stability Stronger anti-GBM activity of actively targeted micelles in vitro Considerably increased tumor repression and survival rates with targeted micelles in vivo in comparison to free Sorafenib and non-targeted controls [154]
PEGylated bilirubin NPs	Cediranib and paclitaxel; D-T7 peptide	Cediranib NPs: 71.5 nm Paclitaxel NPs: 77.2 nm	C6 cell line C6 glioma bearing mice	Significantly more penetration effect, cytotoxicity and median survival time of NPs modified with D-T7 peptide than that of unmodified NPs and saline group [165]
Liposomes composed of DOTAP, DOPE, Cholesterol and DSPE-PEG2000	Doxorubicin and Erlotinib; Transferrin, cell penetrating peptide (PFVYLI)	161.90 ± 4.60 nm	U87, bEnd.3 and glial cell lines	Efficient internalization of drugs in U87, bEnd.3, and glial cells by liposomes Highest apoptotic effect in U87 cells when surface modified with transferrin and CPP [145]
Passively targeted nanoparticles				
Lipid nanocapsules	Sorafenib	54 ± 1 nm	U87MG cell line Orthotopic U87MG glioblastoma model	Reduced cell viability with nanocapsules in vitro Reduced proportion of tumor proliferating cells in vivo Induction of early tumor vascular normalization in vivo [164]
Poly(styrene-co-maleic acid) micelles	Crizotinib and Dasatinib	Crizotinib micelles: 121 nm Dasatinib micelles: 89.14 nm	Various cell lines; U87 and NZG1003 3D spheroids; Female C57BL/6 mice, inoculated with GL261 GBM tumor mass SC	Combination of crizotinib and dasatinib micelles effective in GBM cell lines, 3D spheroids and in in vitro model of angiogenesis and vascular mimicry Combination also effective in in vivo model [156]
Human serum albumin nanoparticles	Ibrutinib and Hydroxychloroquine	160.1 ± 0.7 nm	C6-luc cells Orthotopic glioma xenograft developed by intracranial transplantation of C6-luc cells in mice	Albumin nanoparticles for co-delivery of ibrutinib and hydroxychloroquine showed maximal cytotoxicity to C6 cells Significant prolongation of survival time in in vivo mice model [158]

Abbreviations: DOPE: 1,2-Dioleoyl-sn-glycero-3-phosphoethanolamine; DOTAP: 1,2-Dioleoyl-3-trimethylammonium-propane chloride; DSPE: distearoylphosphatidyl-ethanolamine; GBM: glioblastoma multiforme; IV: intravenous; NP: nanoparticle; PDGFR: platelet-derived growth factor receptor; PEG: poly(ethylene glycol); PLGA: poly-lactide-co-glycolic acid.

## 8. Conclusions and Perspectives

Despite developments in treatments for other cancers, GBM remains a deadly and aggressive disease with poor survival rates. Tumor location, presence of BBB, infiltrative nature, molecular, genetic, and phenotypic heterogeneity, and resistance to chemotherapy, make GBM very difficult to treat [77]. Prognosis and survival of gliomas directly correlates with the ability to be resected surgically [166]. However, complete surgical resection of glioblastomas is not attainable because it is infiltrating and grows into the normal brain tissues. There is a need for novel targeted treatments with prolonged anti-tumor effect, minimum toxicity, and the ability to prevent relapse. TKIs alone or in combination may have great potential for treatment of GBM; however, effectiveness of TKIs is mainly limited by presence of BBB. To achieve optimum delivery of TKIs to brain tumors, different drug delivery methods need to be used. These delivery methods can include nanoparticles, focused ultrasound, CED, intranasal delivery, implantable drug release systems, or intra-arterial drug delivery. Combining targeted therapies with novel drug delivery approaches may enhance their efficacy in GBM, but innovative strategy is required to overcome the drug delivery barriers.

Among the delivery strategies to cross or bypass the BBB, nanocarriers are an example of a promising formulation strategy that can be exploited to load drugs and improve their metabolic stability, intracellular tumor cell delivery and, hence efficacy. Nanoparticles can also provide an opportunity to load multiple drugs for combination therapy and hold great potential for drug delivery via intranasal route. Chemotherapeutic nanomedicine, especially the actively targeted and multifunctional nanocarriers, can address most of the challenges associated with GBM drug delivery. However, more knowledge is required on toxicological profiles, long-term stability, clearance mechanisms, and safety of these therapies. The clinical translation of these therapies is slow and there are many challenges related to commercial development of nanoparticles [143].

Further, different molecular pathways are activated in GBM simultaneously and brain tumors work as an ecosystem with interconnected networks and thus, combination of drugs targeting different vital pathways can prove to be more effective [76,77].

There are many options that can be utilized for achieving success with TKIs for GBM treatment. Future research can focus on exploiting drug delivery strategies for more effective delivery and release of TKIs at tumor site and testing either combination of different TKIs or combination of TKIs with other chemotherapeutic agents acting on different pathways.

## Figures and Tables

**Figure 1 pharmaceutics-15-00059-f001:**
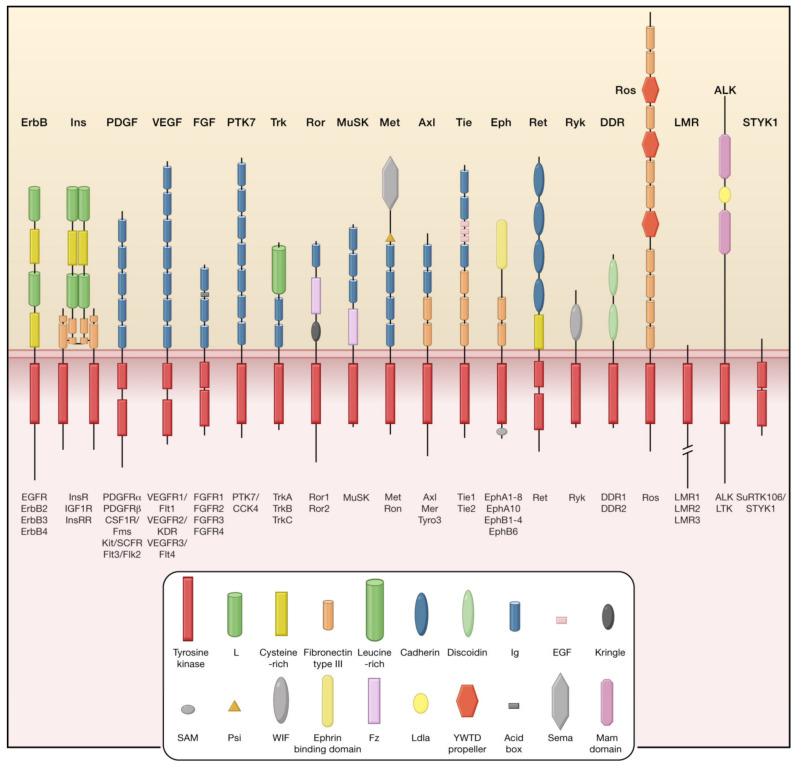
Structure of receptor tyrosine kinases depicting the extracellular portion that binds ligands and a cytoplasmic portion (red rectangle) with tyrosine kinase catalytic activity. Reprinted with permission from reference [36]. Abbreviations: ALK: anaplastic lymphoma kinase; DDR: discoidin domain receptors; EGFR: epidermal growth factor receptor; Eph: erythropoietin-producing hepatocellular carcinoma; FGF: fibroblast growth factor; IGF: insulin-like growth factor; InsR: insulin receptor; Met: mesenchymal-epithelial transition factor; MuSK: musclespecific kinase; PDGF: platelet-derived growth factor; SAM: sterile alpha motif; trk: tropomyosin receptor kinases; VEGF: vascular endothelial growth factor.

**Figure 2 pharmaceutics-15-00059-f002:**
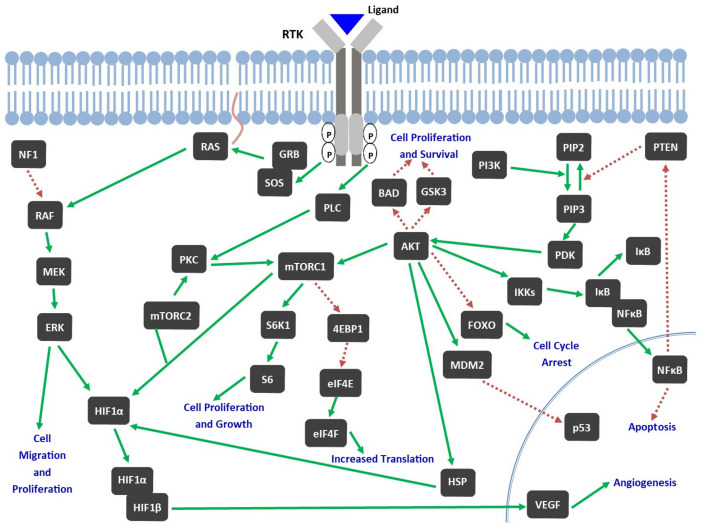
Receptor tyrosine kinase and resultant downstream signaling. Green arrows depict activation and red dashed arrows depict inhibition [13]. Abbreviations: BAD: Bcl2-associated death promoter; EBP: enhancer-binding protein; ERK: extracellular-signal-regulated protein kinase; FOXO: forkhead box O; GSK: glycogen synthase kinase; HIF: hypoxia inducible transcription factors; HSP: heat shock protein; IKK: IκB kinase; MDM2: murine double minute 2; MEK: mitogen-activated protein kinase; mTOR: mammalian target of rapamycin; NF1: neurofibromin 1; NFκB: nuclear factor κB; PDK: phosphoinositide-dependent kinase; PI3K: Phosphatidylionositol-3-kinase; PIP2: phosphatidylinositol bisphosphate; PIP3: phosphatidylinositol 3,4,5-triphosphate; PKC: protein kinase C; PLC: phospholipase C; PTEN: Phosphatase and tensin homolog; RAF: rapidly accelerated fibrosarcoma; RTK: receptor tyrosine kinase; VEGF: vascular endothelial growth factor.

**Figure 3 pharmaceutics-15-00059-f003:**
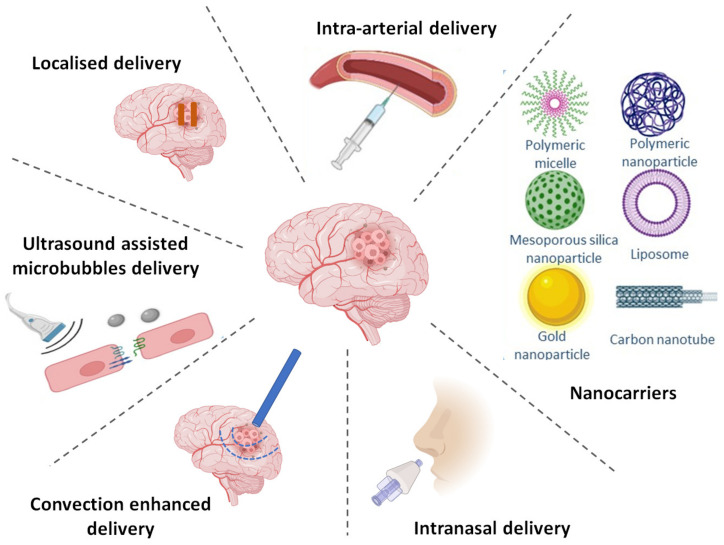
Drug delivery approaches for targeted delivery to brain. Created with BioRender.com.

**Table 1 pharmaceutics-15-00059-t001:** Clinical trials of TKIs in adult high-grade gliomas and their outcomes (ClinicalTrials.gov, accessed on 16 December 2022).

Drug (Dosing)	Clinical Indication and Target of TKI	Clinical Trial	Outcomes of Trial (References)
Imatinib (600 mg/day)	Chronic myeloid leukemia, gastrointestinal tumor Bcr-Abl, KIT, PDGFR	Phase II trial for primary inoperable or incompletely resected and recurrent GBM	No measureable activity Median PFS in newly diagnosed GBM—2.8 months Median PFS in recurrent GBM—2.1 months Major grade 3 AEs—seizure, pneumonia, and vigilance decrease [81]
Imatinib mesylate (oral dose of 600 mg/day) in combination with hydroxyurea (oral dose of 500 mg twice daily) vs. hydroxyurea alone (500 mg 3 times daily)	-	Phase III study in patients with temozolomide resistant progressive glioblastoma	Serious AEs in combination arm: 54.24% Serious AEs in hydroxyurea arm: 38.98% (NCT00154375, ClinicalTrials.gov)
Gefitinib (initial oral dose of 500 mg/day, escalated to 750 mg and then 1000 mg in case patient received enzyme-inducing drugs or dexamethasone)	Non-small cell lung cancer EGFR	Phase II trial for GBM at first recurrence	Tolerable and modest activity 56.6% of the patients suffered therapy failure within the initial 8-week assessment period Most common toxicities—rash and diarrhea [82]
Cediranib (oral dose of 30 mg/day) and gefitinib (oral dose of 500 mg/day) vs. cediranib and placebo	-	Phase II trial for recurrent or progressive GBM	Median PFS—3.6 months (cediranib + gefitinib) and 2.8 months (cediranib + placebo) Median survival times—7.2 months (cediranib + gefitinib) and 5.5 months (cediranib + placebo) Most frequent AEs—fatigue, hypertension, and lymphopenia [79]
Dovitinib (oral dose of 500 mg/day for 5 days, 2 days off weekly on a 28-day cycle)	FGFR, VEGFR, PDGFR	Phase II trial for relapsed or progressive GBM	No efficacy Grade 4/3 toxicities—mainly elevated lipids/lipase, thromboembolic events, fatigue, hypertension, lymphopenia [83]
Sunitinib (oral dose of 37.5 mg/day to start, escalation to 50 mg/day or reduction to 25 or 12.5 mg/day depending on the toxicities)	Gastrointestinal tumor, renal cell carcinoma, pancreatic neuroendocrine tumor KIT, PDGFR, VEGFR1-2, FLT3	Phase II trial for first recurrence of primary GBM	Minimal activity Common toxicities—fatigue, mucositis, dermatitis, gastrointestinal symptoms, dysesthesias, cognitive impairment, thrombocytopenia, and leukocytopenia [84]
Nintedanib (oral dose of 200 mg twice a day)	PDGFR, FGFR, VEGFR	Phase II trial for recurrent high-grade gliomas	Not found to be active for treatment Common AEs—elevated ALT levels, vomiting, abdominal pain, mild diarrhea, nausea [85]
Cabozantinib (Starting oral dose of 140 mg/day considered to be high and then reduced to 100 mg/day)	Progressive metastatic medullary thyroid cancer, renal cell carcinoma, hepatocellular carcinoma previously treated with sorafenib VEGFR1-2, Met, ROS1, RET, AXL, NTRK, KIT	Phase II trial for recurrent or refractory GBM naïve to prior antiangiogenic therapy	Objective response rate—17.6% (dose 140 mg/day) and 14.5% (dose 100 mg/day) Detectable reduction in tumor volume in 90% of patients Common AEs—fatigue, diarrhea, decreased appetite, palmar-plantar erythrodysesthesia, nausea, headache, constipation, hypertension, weight decrease, dysphonia [86]
Cediranib (Initial treatment with 45 mg/day, followed by stepwise dose reduction in patients with dose-limiting toxicities)	VEGFR1-3, KIT, PDGFR	Phase II trial for recurrent GBM	Radiographic partial response—56.7% patients Monotherapy active against recurrent GBM Dose reduction or discontinuation of steroids Grade 3/4 AEs—hypertension, diarrhea, fatigue [87]
Erlotinib (oral dose of 150 mg/day for patients not on drugs that increase CYP3A4 activity and 300 mg/day for patients on drugs that increase CYP3A4 activity, followed by dose escalation)	Non-small cell lung cancer EGFR	Phase II trial for first relapse GBM	Objective response rate—8.3% PFS6—20%, stable disease—33% Common AEs—rash, diarrhea, fatigue, dry skin, headache, exfoliative dermatitis, and nausea [88]
Pazopanib (oral dose of 800 mg/day)	Renal cell carcinoma, soft tissue sarcoma VEGFR, PDGFR, KIT	Phase II trial for recurrent GBM	No prolongation of PFS but in situ biological activity seen according to radiographic responses Median PFS—12 weeks Partial radiographic response seen in 2 patients Decreased contrast enhancement, vasogenic edema, <50% reduction in tumor seen in 9 patients Grade 3/4 AEs—leukopenia, lymphopenia, thrombocytopenia, ALT and AST elevation, hemorrhage, fatigue, and thrombotic events [89]
Cediranib (oral dose of 30 mg/day) monotherapy and cediranib (oral dose of 20 mg/day) combination with lomustine (oral dose of 110 mg/m^2^ once every 6 weeks) versus lomustine alone	VEGFR1-3, KIT, PDGFR	Phase III trial for recurrent GBM	No significant prolongation of PFS with cediranib alone or in combination with lomustine as compared to lomustine alone Cediranib delayed the time to neurologic deterioration and significantly reduced corticosteroid usage Most common AE—diarrhea ≥ grade 3 AEs more frequent in cediranib plus lomustine arm [90]
Dacomitinib (oral dose of 45 mg/day)	Non-small cell lung cancer EGFR, HER2	Phase II trial for recurrent GBM and EGFR amplification with or without variant III (EGFRvIII) deletion	Minimal activity Progressive disease—61.2% Complete response—2% Partial response—4.1% AEs—rash, diarrhea, asthenia, nausea [91]
Afatinib (initiated at 20 mg/day and escalated to 40 and 50 mg/day) with or without temozolomide (75 mg/m^2^ for 21 days every 28-day cycle) vs. temozolomide monotherapy	Non-small cell lung cancer, squamous cell carcinoma of lung EGFR, HER2	Phase I/II trial for recurrent GBM	PFS6—3% (afatinib), 10% (afatinib + TMZ), 23% (TMZ) Median PFS longer in afatinib treated patients with EGFRvIII-positive tumors than EGFRvIII-negative tumors More frequent AEs in patients treated with afatinib and combination Most frequent phase II AE—diarrhea and rash [92]
Bevacizumab (5 mg/kg intravenously every 2 weeks) alone and in combination with **sorafenib** (200 mg twice a day for 1–5 days a week then modified to 200 mg/day because of toxicities)	Renal cell carcinoma, hepatocellular carcinoma, differentiated thyroid cancer VEGFR1-3, TIE2, PDGFR, FGFR, BRAF, CRAF, KIT, FLT-3	Phase II trial for recurrent GBM	No improvement in outcome of patients with the combination as compared to bevacizumab alone [93]
Axitinib (treatment initiated at oral dose of 5 mg twice daily and adjusted according to toxicity) or bevacizumab or lomustine	Renal cell carcinoma VEGFR1-3, PDGFR, KIT, FLT-3	Phase II trial for recurrent GBM	PFS6—34% (axitinib), 28% (control) Axitinib monotherapy displayed clinical activity and manageable toxicity Axitinib treated patients developed hypoalbuminemia, hypertension, increased hemoglobin, diarrhea, oral hypersensitivity, dysphonia, hypothyroidism, fatigue, and rash more often [94]
Axitinib (started at oral dose of 5 mg twice daily)Avelumab (10 mg/kg intravenously over 60 min every 2 weeks)	-	Phase II trial for recurrent GBM	Complete or partial response in 27.8% patients Common AEs—Dysphonia, lymphopenia, diarrhea, arterial hypertension [95]
Radiation plus temozolomide with or without vandetanib (100 mg/day 5–7 days prior to radiation)	Unresectable or metastatic medullary thyroid cancer EGFR, VEGFR2-3, RET	Phase II trial for newly diagnosed GBM	Insignificant prolongation of overall survival by addition of vandetanib to radiation plus temozolomide [96]

Clinical indication and target of TKIs from reference Huang et al. [15]. Abbreviations: AE: adverse event; ALT: alanine transaminase; EGFR: epidermal growth factor receptor; FGFR: fibroblast growth factor receptor; GBM: glioblastoma multiforme; PDGFR: platelet-derived growth factor receptor; PFS: progression free survival; PFS6: PFS at 6 months; VEGFR: vascular endothelial growth factor receptor.

## Data Availability

Not applicable.

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
