# Peer review of "Tyrosine Kinase Inhibitors for Glioblastoma Multiforme: Challenges and Opportunities for Drug Delivery"

_pharmaceutics, 2022, doi:10.3390/pharmaceutics15010059_

Round 1

Reviewer 1 Report

The review highlighs the role of tyrosine kinases in the pathophysiology  of Glioblastoma multiforme and Tyrosine kinase inhibitors use for Glioblastoma multiforme treatment. It also highlights the limitations for clinical trials utilizing Tyrosine kinase inhibitors and potential promising drug delivery strategies for targeted delivery to brain tumors.

-In table 1: the dosing/regimen of drug administration should be indicated for each clinical trial

-Although clincial trials utilied chemical inhibitors of tyrosine kinases, the authors should highlight the reported natural tyrosine kinase inhibitors 

-A paragraph summarizing experimental models used for induction of Glioblastoma multiforme can be added 

-Conclusion and prespectives should be summarized and be more focused

Author Response

  • The dosing regimen of drug administration has been included for each clinical trial in table1.
  • A paragraph about natural tyrosine kinase inhibitors has been included in section 4 and a review article has been cited, which provides the details regarding reported natural products as inhibitors of different tyrosine kinases.
  • A section (section 3) on preclinical glioblastoma models has been included.
  • Conclusions and perspectives have been accordingly summarized.

Reviewer 2 Report

The manuscript exhibited valuable information about TKI treatment for GBM. The authors should pay attention to the following points to let their work more scientific.

1.     In the first paragraph of “introduction”section, the classification of gliomas has been modified. I’d like the authors to cite the current criteria for grading. 

2.     In section 2.1.2, IDH2 mutations also can be found in GBM. Also, H3K27M is an important genetic abnormality in GBM.

3.     In section 6, I do not think “Chemotherapy is an ‘effective’ strategy for (GBM) treatment”. But, it is a strategy without optional alternatives

4.     In section 6, there are many examples that should be removed and discussed in other sections. This section should be described more precisely.

5.     Can the authors add some words about children's GBM and the clinical trial results of TKI if possible?

Author Response

  • The classification of glioblastoma in the revised manuscript (Introduction) has been modified according to the 2021 WHO classification of tumors of the CNS in the introduction.
  • IDH2 (section 2.1.2) and H3K27M (section 2.1.4) mutations in glioblastoma have been included.
  • The phrase 'Chemotherapy is an effective strategy for GBM treatment' has been removed from the section 'Conclusions and perspectives'.
  • Conclusions and perspectives have been described more precisely, and a few examples from there have been moved to section 5.
  • A section 6 has been added on paediatric high-grade gliomas, and a review article providing details about clinical trials of tyrosine kinase inhibitors in paediatric tumors has been cited.

Reviewer 3 Report

This manuscript is a well-written summary of the current situation on GBM treatment possibilities and related issues. Some ideas:
ad 2.1: Listing of mutated genes could involve IDH2 a.o. in terms of comprehensiveness. One could also align the intro to WHO classification of CNS tumors 2021 and refer to the left-out subgroup of IDH-mutant GBMs.
ad Table 1: could indicate (if repurposed) former utilization of drug (cancer type) and target (gene) if applicable.

Author Response

  • IDH2 mutation has been included in section 2.1.2, and the introduction has been modified to include the 2021 WHO classification of tumors of the CNS.
  • The clinical indication of the FDA-approved tyrosine kinase inhibitors and the target genes of tyrosine kinase inhibitors has been included in table 1.

Round 2

Reviewer 1 Report

The authors addressed my comments and the review has been markedly improved.